# Multi-Time-Scale Energy Storage Optimization Configuration for Power Balance in Distribution Systems

Qiuyu Lu [1], Xiaoman Zhang [2], Yinguo Yang [1], Qianwen Hu [2], Guobing Wu [1], Yuxiong Huang [2,*][iD], Yang Liu [1] and Gengfeng Li [2]

[1] Guangdong Power Grid Dispatch and Control Center, Guangzhou 510600, China; luqiuyu0708@gdpc.gd.csg.cn (Q.L.); yangyinguo@gddd.csg.cn (Y.Y.); wuguobing@gdpc.gd.csg.cn (G.W.); liuyang@gdpc.gd.csg.cn (Y.L.)

[2] State Key Laboratory of Electrical Insulation and Power Equipment, Xi'an Jiaotong University, Xi'an 710049, China; zxm2022@stu.xjtu.edu.cn (X.Z.); qwhu089@stu.xjtu.edu.cn (Q.H.); gengfengli@xjtu.edu.cn (G.L.)

\* Correspondence: yuxionghuang@xjtu.edu.cn

**Abstract:** As the adoption of renewable energy sources grows, ensuring a stable power balance across various time frames has become a central challenge for modern power systems. In line with the "dual carbon" objectives and the seamless integration of renewable energy sources, harnessing the advantages of various energy storage resources and coordinating the operation of long-term and short-term storage have become pivotal directions for future energy storage deployment. To address the complexities arising from the coupling of different time scales in optimizing energy storage capacity, this paper proposes a method for energy storage planning that accounts for power imbalance risks across multiple time scales. Initially, the Seasonal and Trend decomposition using the Loess (STL) decomposition method is utilized to temporally decouple actual operational data. Subsequently, power balance computations are performed based on the obtained data at various time scales to optimize the allocation of different types of energy storage capacities and assess the associated imbalance risks. Finally, the effectiveness of the proposed approach is validated through hourly applications using real-world data from a province in southern China over recent years.

**Keywords:** energy storage planning; renewable energy integration; STL decomposition method

## 1. Introduction

With the continuous advancement of electrification, energy has become the primary battleground for mitigating global climate change [1]. Electricity serves as the vanguard in driving the development of a low-carbon society. Promoting the construction of a new type of power system, with wind and solar power as the main components, is a crucial pathway for energy conservation and emission reduction. However, the output of renewable energy sources such as wind and solar power is significantly influenced by weather changes, posing considerable challenges due to their intermittency and volatility [2]. Furthermore, mismatches between renewable energy generation and demand at different scales can also affect electricity supply, leading to power imbalances [3]. This is manifested in various aspects, such as fluctuations in electricity prices and power rationing [4]. In 2021, natural gas prices in Europe experienced significant fluctuations, rising by over 600% due to extreme weather conditions and uncertainty surrounding renewable energy output. Consequently, the average electricity price in major European countries has surpassed €300 per megawatt-hour, reaching historic highs [5]. In September 2021, power rationing was "forced" in the three northeastern provinces of China due to significant supply–demand gaps, impacting the lives of residents and societal production.

To address the challenges facing the construction of new power systems and the seasonal imbalances between renewable energy and demand [6], and to mitigate the drastic

fluctuations in electricity prices and occurrences of power rationing, several measures need to be taken [7,8]. On the one hand, it is essential to strategically plan for flexible resources like energy storage, which exhibit temporal and spatial transfer characteristics, while also maintaining the adequate capacity of traditional energy generation [9,10]. On the other hand, ensuring the economic viability of energy storage resource allocation is crucial alongside ensuring the reliability of grid operations [11,12]. Efficient utilization of various types of energy storage resources to address energy imbalances is vital [13,14]. This involves swiftly adjusting and activating energy storage systems during fluctuations in renewable energy output to ensure a stable electricity supply, thereby facilitating energy transition and promoting the development of renewable energy [15,16]. Therefore, it is imperative to strategically plan energy storage resources, leveraging the unique characteristics of different types of storage to tackle the imbalance issues in power systems [17,18].

Current research by experts and scholars has extensively addressed the issue of seasonal imbalance in electricity supply. Article [19] developed a coordinated optimization model for generation–grid–storage systems, incorporating a comprehensive yearly, hourly operational simulation and utilizing a compact panoramic time series to expedite model solving. Article [20] proposed an energy storage planning model that considers the seasonal imbalances resulting from the long-term uncertainty of renewable energy generation, yet it did not account for the impacts of short-term electricity fluctuations. However, there is a lack of research specifically addressing the imbalance risks arising from both the long-term and short-term uncertainties in electricity supply. Articles [21–23] integrated renewable energy with ammonia production, presenting a planning approach that considers renewable energy uncertainty, ammonia storage, and renewable energy generation. Article [24] combined renewable energy generation with flexible resources like thermal power generation, establishing an optimization model for generating combinations to minimize load loss. Article [25] optimized energy storage in regional energy internet based on user energy demands and future load trends, facilitating multi-energy coordination. Nevertheless, fewer studies have focused on the coordinated integration of multiple renewable energy sources with energy storage and other flexible resources [26].

Addressing the aforementioned shortcomings, this paper proposes an energy storage planning method that considers power imbalance risks across multiple time scales. Based on the collection of actual operational data from a specific province, a decomposition method is employed to temporally decouple the output of renewable energy sources and load profiles, thereby obtaining seasonal and periodic components of both renewable energy output and load variations. Utilizing the obtained decomposition results and considering the characteristics of different types of energy storage, hydrogen storage suitable for long-term storage and electrochemical storage capable of rapid charge and discharge are selected as different types of energy storage technologies in the planning process to meet the energy system's demands across different time scales. In the process of energy storage planning, the marginal costs of energy storage construction are taken into account to optimize energy storage planning decisions, maximizing resource utilization efficiency and economic benefits. The main contributions of this paper are summarized as follows:

- Considering the inclusion of marginal costs in energy storage cost calculations to optimize the relationship between energy storage capacity and storage costs;
- Addressing the characteristics of changes in renewable energy and load profiles with economic development and seasonal variations in the new power system, utilizing a hybrid energy storage technology combining hydrogen storage and chemical energy storage to achieve supply–demand balance;
- Employing Seasonal–Trend decomposition using LOESS (STL) decomposition technology to analyze and decompose data at long time scales, enabling the derivation of regional energy storage deployment schemes.

The remaining sections of the article are organized as follows: Section 2 provides an overview of the overall methodology and approach adopted in the study. Section 3

elaborates on the mathematical model used for energy storage planning. Section 4 presents the optimization configuration of energy storage resources for a specific region based on recent operational data of wind power, solar power, and load profiles. This chapter integrates the proposed model to offer an optimized allocation plan for energy storage resources in the region.

## 2. Methodology

### 2.1. Energy Storage Type Selection

In the selection of energy storage types, this paper adopts hydrogen storage and electrochemical storage as two energy storage technologies [27], which are, respectively, used to balance the long-term uncertainty and short-term uncertainty in the renewable energy system [28,29]. In addition, hydrogen can be obtained through clean energy power electrolysis technology. The obtained hydrogen is stored in efficient hydrogen storage devices, and then, using fuel cell technology, the stored energy is fed back to the grid. On the other hand, hydrogen has high energy density, relatively low operation and maintenance costs, can be stored for a long time without losing too much energy, and there is no self-discharge phenomenon [30,31]. Therefore, it is suitable for large-scale long-term storage to cope with the seasonal imbalance caused by fluctuations in new energy output or the intermittency of solar and wind power generation, thereby alleviating the long-term uncertainty of the system.

In contrast, electrochemical energy storage technologies such as lithium-ion batteries and sodium-sulfur batteries offer advantages such as rapid response times and high energy densities, enabling the quick release of stored energy over short durations. However, they have a limited life cycle, higher safety risks, and are relatively expensive. As a result, they are primarily deployed in scenarios requiring grid peak shaving and frequency regulation, particularly suited to address short-term load fluctuations or unpredictable energy supply variations in the system, thereby mitigating short-term uncertainty.

In conclusion, hydrogen storage and electrochemical energy storage offer distinct solutions for addressing the long-term uncertainty and short-term uncertainty in renewable energy systems, respectively. Together, they can provide support for the balance and stability of energy systems.

### 2.2. Energy Storage Planning Methodology

The objective of this paper is to utilize the temporal and spatial transfer characteristics of different types of energy storage to mitigate the risk of power imbalance in the new power system, thereby achieving the core goal of clean and low-carbon electricity and promoting the development of green power.

Figure 1 illustrates the flow of energy in the new power system. The primary sources of energy mainly include solar power and wind power. Energy storage predominantly occurs through hydrogen storage and electrochemical energy storage, while energy is consumed across various types of electrical load demand systems.

Figure 2 depicts the overall flowchart of optimizing energy storage planning, divided into four steps. Firstly, obtain the historical operational data of the system, including wind power, solar power, and load data for all 8760 h of the year. Secondly, the collected data from Step 1 are processed to calculate the net load of the system. Apply the STL decomposition method to decompose the net load data into trend, seasonal, and residual components. Compute and validate the trendiness and seasonality of the obtained data. Next, an optimization model aimed at minimizing the overall operating cost of the system is constructed, with constraints including power balance constraints and maximum load shedding constraints. Finally, we use the Gurobi solver to optimize hydrogen and electrochemical storage capacities, along with the power output for each time period.

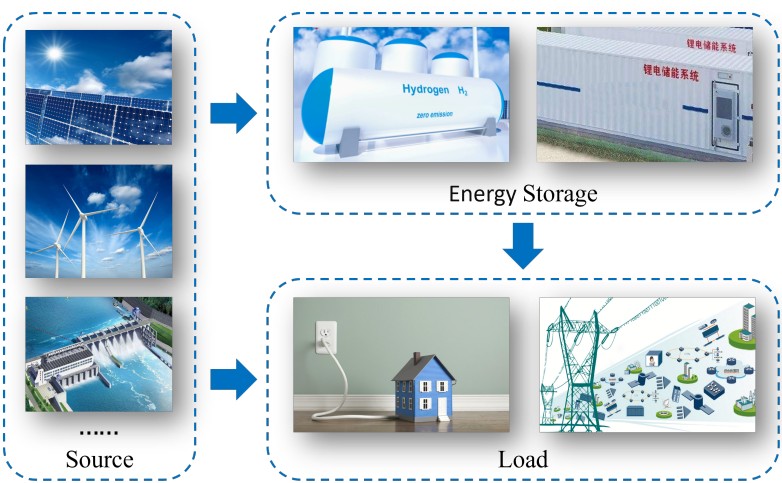

**Figure 1.** Energy flow in distribution systems.

---

**Step 1: Obtain historical system operation data.**
Include wind power, solar power, and load data for all 8760 hours of the year.

↓

**Step 2: Data processing .**
Calculate the net load of the system based on the collected data. Decompose the net load data into trend, seasonal, and residual components. Then the trendiness and seasonality of the data will be computed and validated.

↓

**Step 3: Construct an optimization model**
The objective is minimizing the overall operating cost of the system. The constraints include power balance constraints, maximum load shedding constraints, maximum curtailment of wind and solar constraints, battery operation constraints, and so on.

↓

**Step 4: Utilize the Gurobi solver to solve the optimization problem**
Obtain the optimized capacities for hydrogen storage and electrochemical storage, as well as the power output of storage for each time period.

---

**Figure 2.** Diagram of the article flow.

### 2.3. STL Decomposition Method

Seasonal–Trend decomposition using LOESS (STL) [32] is a robust method for decomposing time series data based on the additive principle [33]. This decomposition method was proposed by R. B. Cleveland, W. S. Cleveland, McRae, and Terpenning in 1990. Its distinguishing feature is its ability to obtain stable trends and seasonal components, with strong resistance to short-term anomalies in the data. Due to the strong seasonality and regularity of renewable energy output such as photovoltaic and wind power, it is possible to decompose the data of renewable energy output and load data based on historical data. Assuming the net load data are a time series $y(t)$, and assuming they follow an additive model, they can be decomposed as follows:

$$y(t) = S(t) + T(t) + R(t) \tag{1}$$

In this equation, $T(t)$ represents the trend value at time $t$; it represents the part of the net load component that exhibits long-term fluctuations and is predictable. $S(t)$ denotes the seasonal component at time $t$; it represents the part that exhibits short-term fluctuations and is predictable. And $R(t)$ stands for the residual component at time $t$; it represents the part that exhibits short-term fluctuations but is not predictable.

The decomposition mainly consists of two parts: the inner loop and the outer loop. The main function of the inner loop is to perform trend fitting and calculate the periodic components, while the main function of the outer loop is to adjust the weights of robust optimization. The process is illustrated in Table 1, as shown in the decomposition flowchart.

**Table 1.** STL decomposition process.

| |
|---|
| **Outer Loop** |
|     Calculate weights |
| **Inner Loop** |
|     Step 1: detrend; |
|     Step 2: smooth periodic subsequence; |
|     Step 3: low-pass filtering of periodic subsequences; |
|     Step 4: remove trend from smoothed periodic subsequences; |
|     Step 5: detrending; |
|     Step 6: trend smoothing. |

To verify the validity of using STL decomposition, it is necessary to calculate the trend and seasonality of the selected data. The trend reflects the long-term fluctuations and predictability of the net load data, while the seasonality reflects the short-term fluctuations and predictability. To compute the trend and seasonality, the variances of the trend component, seasonality component, and residual component need to be calculated separately. Subsequently, calculate the strength of the trend and seasonality based on Equations (2) and (3).

The strength of the trend can be defined as:

$$F_{\mathrm{T}} = \max(0, 1 - \frac{Var(R(t))}{Var(T(t) + R(t))}) \tag{2}$$

Hence, the strength of the trend lies between 0–1 and $F_{\mathrm{T}}$, where a value closer to 0 indicates that the sequence has almost no trend, and a value closer to 1 indicates a stronger trend intensity in the time series.

Similarly, the strength of seasonality can be defined as:

$$F_{\mathrm{S}} = \max(0, 1 - \frac{Var(R(t))}{Var(S(t) + R(t))}) \tag{3}$$

The strength of seasonality lies between 0–1 and $F_{\mathrm{S}}$, where a larger $F_{\mathrm{S}}$ value corresponds to a stronger seasonality intensity.

## 3. Mathematical Model

The established model is based on the following assumptions:

- Only photovoltaic and other renewable energy devices, as well as flexible resources such as energy storage, participate in the operation balance of the power system;
- Considering the insufficiency of renewable energy output to meet the system load demand, the addition of hydrogen energy from alternative sources to compensate for the imbalance in power is being considered;
- Only consider hydrogen obtained through clean energy sources and fed back to the grid through fuel cell technology. The model does not consider the demand for hydrogen energy by the hydrogen industry chain.

### 3.1. Objective Function

The model aims to minimize the total cost and establishes the objective function as shown in Formula (4). The total cost of the power system includes two parts: the investment cost of energy storage construction and the operational cost of the system. The unit for costs is in RMB (Chinese Yuan), and the unit for capacity is in kWh (kilowatt-hours).

$$\min C = C^{\mathrm{INV}} + C^{\mathrm{OPE}} \tag{4}$$

where $C$ represents the total cost of the grid, $C^{\mathrm{INV}}$ represents the investment cost of the grid, and $C^{\mathrm{OPE}}$ represents the operational cost of the grid.

Formulas (5)–(10) demonstrate the investment cost of energy storage and related equations.

$$C^{\text{INV}} = (C^{\text{INV\_che}} + C^{\text{INV\_hyd}}) \times CRF \tag{5}$$

where $C^{\text{INV\_che}}$ represents the upfront cost of electrochemical energy storage, $C^{\text{INV\_hyd}}$ represents the upfront cost of hydrogen energy storage. $CRF$ represents the capital recovery factor.

$$CRF = \frac{\gamma(1+\gamma)^n}{((1+\gamma)^n - 1)} \tag{6}$$

In the formula, $\gamma$ represents the annual interest rate of investment cost and $n$ represents the total lifespan of energy storage.

Considering the capacity constraints of electrochemical energy storage and hydrogen storage, Formulas (7) and (8) incorporate the calculation of marginal costs when calculating the construction cost of energy storage, aiming to optimize the capacity allocation of energy storage.

$$C^{\text{INV\_che}} = c_{\text{mc,che}} cap_{\text{pre,che}} + c_{\text{unit,che}} N_{\text{unit,che}} \tag{7}$$

where $c_{\text{mc,che}}$ represents the marginal cost of electrochemical energy storage. $cap_{\text{pre,che}}$ represents the capacity of pre-installed electrochemical energy storage. $c_{\text{unit,che}}$ represents the cost per unit of electrochemical energy storage. $N_{\text{unit,che}}$ represents the number of pre-installed electrochemical energy storage units.

$$C^{\text{INV\_hyd}} = c_{\text{mc,hyd}} cap_{\text{pre,hyd}} + c_{\text{unit,hyd}} N_{\text{unit,hyd}} \tag{8}$$

where $c_{\text{mc,hyd}}$ represents the marginal cost of hydrogen energy storage, $cap_{\text{pre,hyd}}$ represents the capacity of pre-installed hydrogen energy storage, $c_{\text{unit,hyd}}$ represents the cost per unit of hydrogen energy storage, $N_{\text{unit,hyd}}$ represents the number of pre-installed hydrogen energy storage units.

Discrete variables are transformed into continuous variables to facilitate subsequent optimization calculations.

$$N_{\text{unit,che}} = \left\lceil \frac{cap_{\text{pre,che}}}{cap_{\text{N,che}}} \right\rceil \tag{9}$$

$$N_{\text{unit,hyd}} = \left\lceil \frac{cap_{\text{pre,hyd}}}{cap_{\text{N,hyd}}} \right\rceil \tag{10}$$

where $cap_{\text{N,che}}$ represents the capacity per unit of electrochemical energy storage, and $cap_{\text{N,hyd}}$ represents the capacity per unit of hydrogen.

The operational cost of the power system and its specific expressions are shown in Equations (11)–(14). The operational cost of the power system consists of four parts: shedding load penalty cost, wind and solar spillage cost, profit from supplying load, and energy storage operation cost.

$$C^{\text{OPE}} = C^{\text{CUT\_LOAD}} + C^{\text{CUT\_NE}} + C^{\text{GET\_NE}} - C^{\text{SUP\_LOAD}} \tag{11}$$

In the equations, $C^{\text{CUT\_LOAD}}$ represents the total cost of shedding the load, $C^{\text{CUT\_NE}}$ represents the total cost of wind and solar spillage, $C^{\text{SUP\_LOAD}}$ represents the total revenue from the power supply, $C^{GET\_NE}$ represents the total consumption of renewable energy.

$$C^{\text{CUT\_LOAD}} = c_{\text{load,cut}} \sum_{d=1}^{D} \sum_{t=1}^{T} E_{\text{load,cut}} \tag{12}$$

where $c_{\text{load,cut}}$ represents the unit cost of shedding the load, $E_{\text{load,cut}}$ represents the total amount of shed load,

$$C^{\text{CUT\_NE}} = c_{\text{ne,cut}} \sum_{d=1}^{D} \sum_{t=1}^{T} E_{\text{ne,cut}} \tag{13}$$

where $c_{\text{ne,cut}}$ represents the unit cost of wind and solar spillage, $E_{\text{ne,cut}}$ represents the total amount of renewable energy,

$$C^{\text{SUP\_LOAD}} = c_{\text{load,get}}(\sum_{d=1}^{D}\sum_{t=1}^{T} cap_{\text{load}} - \sum_{d=1}^{D}\sum_{t=1}^{T} cap_{\text{load,cut}} + \sum_{d=1}^{D}\sum_{t=1}^{T} cap_{\text{ne,cut}}) \tag{14}$$

where $c_{load,get}$ represents the unit revenue from the power supply, and $cap_{\text{load}}$ represents the total load. $cap_{\text{load,cut}}$ represents the total cutting load. $cap_{\text{load,ne}}$ represents the total cutting renewable energy. $D$ represents the total number of days in a year, which is 365, and $d$ represents a specific day within that year. $T$ represents the total number of hours in a day, which is 24, and $t$ represents a specific hour within that day.

### 3.2. Constraint Conditions

The constraints established in this paper include power balance constraints, maximum shedding load constraints, maximum wind and solar spillage constraints, battery operation constraints, etc.

#### 3.2.1. Power Balance Constraint

Formulas (15)−(17) represent the constraints on the power balance of the system, including the balance between long-term energy storage and trend components, the balance between short-term energy storage and seasonal components, and the real-time balance of power during grid operation.

$$P_{d,t}^{\text{che,long}} + P_{d,t}^{\text{hyd,long}} = \Delta P_{d,t}^{\text{ne,trend}}, 1 \leqslant d \leqslant D, 1 \leqslant t \leqslant T \tag{15}$$

where $P_{d,t}^{\text{che,long}}/P_{d,t}^{\text{hyd,long}}$ represents the charging and discharging power of long-term energy storage, and $\Delta P_{d,t}^{\text{ne,trend}}$ represents the trend component.

$$P_{d,t}^{\text{che,short}} + P_{d,t}^{\text{hyd,short}} = \Delta P_{d,t}^{\text{ne,seasonal}} + \Delta P_{d,t}^{\text{ne,resid}}, 1 \leqslant d \leqslant D, 1 \leqslant t \leqslant T \tag{16}$$

$$P_{d,t}^{\text{che}} + P_{d,t}^{\text{hyd}} - P_{d,t}^{\text{load,cut}} \geqslant \Delta P_{d,t}^{\text{ne}} - P_{d,t}^{\text{ne,cut}}, 1 \leqslant d \leqslant D, 1 \leqslant t \leqslant T \tag{17}$$

In the equations, $P_{d,t}^{\text{che,short}}/P_{d,t}^{\text{hyd,short}}$ represents the charging and discharging power of short-term energy storage, and $\Delta P_{d,t}^{\text{ne,seasonal}}$ represents the seasonal component of the net load. $\Delta P_{d,t}^{\text{ne,resid}}$ represents the residual component of the net load. $\Delta P_{d,t}^{\text{ne}}$ represents the power output of renewable energy.

#### 3.2.2. Load Cut Constraint

Formula (18) represents that the maximum shedding load cannot exceed the specified maximum value $P_{d,t}^{\text{load,cut, max}}$.

$$0 \leqslant P_{d,t}^{\text{load,cut}} \leqslant P_{d,t}^{\text{load,cut, max}}, 1 \leqslant d \leqslant D, 1 \leqslant t \leqslant T \tag{18}$$

In the equation, $P_{d,t}^{\text{load,cut}}$ represents the total amount of load shedding. $P_{d,t}^{\text{load,cut, max}}$ represents the maximum load shedding amount.

#### 3.2.3. Wind and Solar Spillage Constraint

Formula (19) represents that the maximum wind and solar spillage at different times must not exceed the given maximum value $P_{d,t}^{\text{ne,cut, max}}$.

$$0 \leqslant P_{d,t}^{\text{ne,cut}} \leqslant P_{d,t}^{\text{ne,cut, max}}, 1 \leqslant d \leqslant D, 1 \leqslant t \leqslant T \tag{19}$$

In the equation, $P_{d,t}^{\text{ne,cut}}$ represents the total amount of wind and solar spillage, and $P_{d,t}^{\text{ne,cut,max}}$ represents the maximum removal amount of renewable energy.

### 3.2.4. Battery Operation Constraint

Formulas (20) and (21) calculate the capacity of energy storage at different times, including self-discharge, charging, and discharging of energy storage.

$$E_{d,t+\Delta t}^{\text{che}} = E_{d,t}^{\text{che}}(1 - \eta_{\text{loss}}^{\text{che}})^{\Delta t} + (P_{d,t}^{\text{che,cha}}\eta_{\text{cha}}^{\text{che}} - P_{d,t}^{\text{che,dis}}/\eta_{\text{dis}}^{\text{che}})\Delta t, 1 \leqslant d \leqslant D, 1 \leqslant t \leqslant T \quad (20)$$

$$E_{d,t+\Delta t}^{\text{hyd}} = E_{d,t}^{\text{hyd}}(1 - \eta_{\text{loss}}^{\text{hyd}})^{\Delta t} + (P_{d,t}^{\text{hyd,cha}}\eta_{\text{cha}}^{\text{hyd}} - P_{d,t}^{\text{hyd,dis}}/\eta_{\text{dis}}^{\text{hyd}})\Delta t, 1 \leqslant d \leqslant D, 1 \leqslant t \leqslant T \quad (21)$$

where $P_{d,t}^{\text{che,cha}}/P_{d,t}^{\text{che,dis}}$ represents the charging and discharging power of electrochemical energy storage, $P_{d,t}^{\text{hyd,cha}}/P_{d,t}^{\text{hyd,dis}}$ represents the charging and discharging power of hydrogen energy storage, and $\eta_{\text{loss}}^{\text{che}}/\eta_{\text{loss}}^{\text{hyd}}$ represents the efficiency of energy storage charging and discharging.

Formulas (22) and (23) constrain the maximum charging and discharging rates of energy storage.

$$0 \leqslant P_{d,t}^{\text{che,cha}}, P_{d,t}^{\text{che,dis}} \leqslant P_{d,t}^{\text{che,max}}, 1 \leqslant d \leqslant D, 1 \leqslant t \leqslant T \quad (22)$$

$$0 \leqslant P_{d,t}^{\text{hyd,cha}}, P_{d,t}^{\text{hyd,dis}} \leqslant P_{d,t}^{\text{hyd,max}}, 1 \leqslant d \leqslant D, 1 \leqslant t \leqslant T \quad (23)$$

where $P_{d,t}^{\text{che,max}}/P_{d,t}^{\text{hyd,max}}$ represents the maximum charging and discharging power of energy storage.

Formulas (24) and (25) restrict the upper and lower bounds of energy storage charging and discharging.

$$E_{d,t}^{\text{che,min}} \leqslant E_{d,t}^{\text{che}} \leqslant E_{d,t}^{\text{che,max}}, 1 \leqslant d \leqslant D, 1 \leqslant t \leqslant T \quad (24)$$

$$E_{d,t}^{\text{hyd,min}} \leqslant E_{d,t}^{\text{hyd}} \leqslant E_{d,t}^{\text{hyd,max}}, 1 \leqslant d \leqslant D, 1 \leqslant t \leqslant T \quad (25)$$

where $E_{d,t}^{\text{che}}/E_{d,t}^{\text{hyd}}$ represents the capacity of energy storage, $E_{d,t}^{\text{che,min}}/E_{d,t}^{\text{hyd,min}}$ represents the minimum capacity allowed for energy storage, $E_{d,t}^{\text{che,max}}/E_{d,t}^{\text{hyd,max}}$ represents the maximum capacity allowed for energy storage.

Formulas (26) and (27) ensure that the energy storage capacity remains constant within a day.

$$E_{d+1,0}^{\text{che}} = E_{d,0}^{\text{che}}, 1 \leqslant d \leqslant D \quad (26)$$

$$E_{d+1,0}^{\text{hyd}} = E_{d,0}^{\text{hyd}}, 1 \leqslant d \leqslant D \quad (27)$$

where $E_{d,0}^{\text{che}}/E_{d,0}^{\text{hyd}}$ represents the initial capacity of energy storage.

## 4. Case Study

Based on the PV, wind power, and load data of a region in southern China for more than one year, the above model is applied for verification. Firstly, the obtained data are organized and analyzed to obtain net load data, and a net load curve is plotted. Secondly, the net load data are decomposed using the decomposition method to obtain trend components, seasonal components, and residual components, and the trend strength and seasonality strength are calculated to verify trend and seasonality. Then, based on the multi-type energy storage planning model in Section 3 and the costs in Table 2, the configuration of hydrogen storage and electrochemical energy storage, as well as the output during each period throughout the year, are obtained. Finally, we validate the effectiveness of the model.

Firstly, based on the operational data of wind power, photovoltaic power, and load for 8760 h in the region, the net load curve is plotted as shown in Figure 3. From the

graph, it can be observed that the load exhibits significant variations throughout the day, characterized by "two peaks and one trough". The "two peaks" occur around 6–9 a.m. and 6–10 p.m., primarily driven by residential, commercial, and industrial electricity consumption. The "trough" occurs around 11 a.m.–4 p.m., mainly due to the substantial generation of wind and solar power. Over the year, the net load shows significant seasonal variations. In winter and spring, the net load is negative, indicating that the generation of renewable energy exceeds electricity consumption. In summer and autumn, the net load is positive, indicating that the generation of renewable energy is lower than electricity consumption.

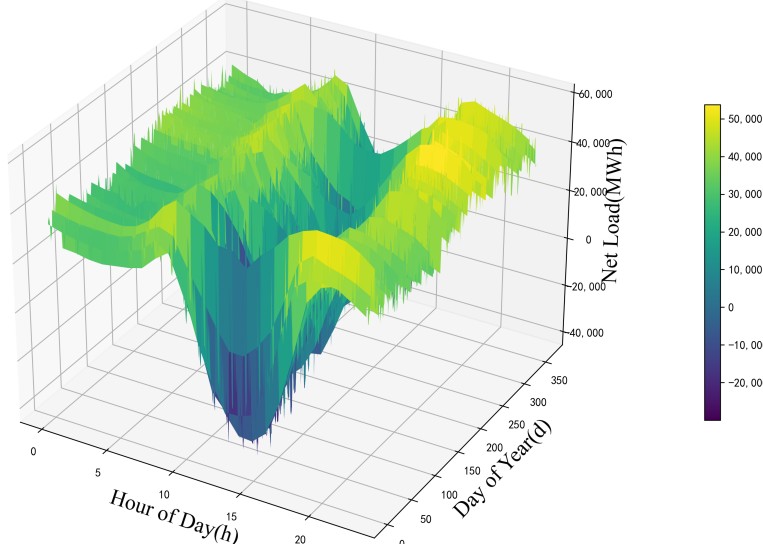

**Figure 3.** Net load curve for 8760 h.

The trend component, seasonal component, and residual component obtained through STL decomposition are shown in Figures 4 and 5. By using Formulas (2) and (3) to calculate the strength of trend and seasonality after decomposition, we obtain $F_T = 0.87154$ and $F_S = 0.96619$, both of which are close to 1. This indicates that the net load data for this location exhibit both trend and seasonality, hence validating the use of the STL decomposition method for calculation.

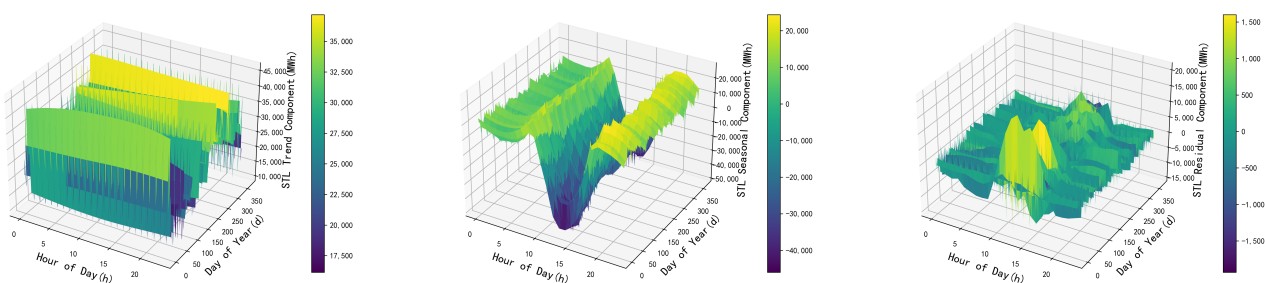

**Figure 4.** Three-dimensional decomposition plot of the net load.

After obtaining the trend component, seasonal component, and residual component of the net load for 8760 h, the multi-type energy storage planning model described in Section 3 and the initial value settings in Table 2 are utilized [34]. Hydrogen energy storage is used to balance the long-term imbalance component of the power system, while electrochemical energy storage, with its rapid charging and discharging properties, is utilized to balance the short-term power imbalance of the power system. This process results in the configuration of hydrogen energy storage and electrochemical energy storage, along with the power output throughout the year at different times.

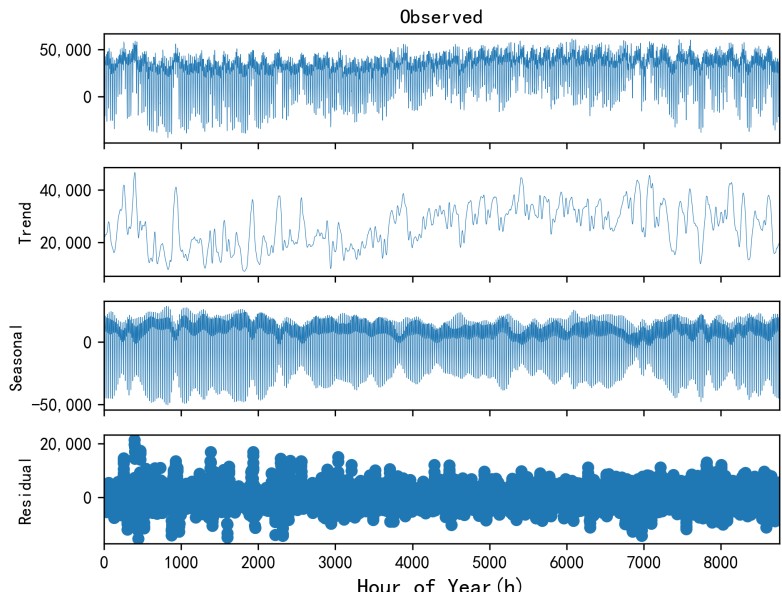

**Figure 5.** Two-dimensional decomposition plot of the net load.

**Table 2.** Reference value settings.

| Name | Numerical Values |
| --- | --- |
| The installation cost of electrochemical energy storage | 1.66 RMB/kWh |
| The installation cost of hydrogen energy storage | 8 RMB/kWh |
| The marginal cost of electrochemical energy storage | 1 RMB/kWh |
| The marginal cost of hydrogen energy storage | 4 RMB/kWh |
| The operating cost of electrochemical energy storage | 0.5 RMB/kWh |
| The operating cost of hydrogen energy storage | 0.1 RMB/kWh |
| The cost of wind and solar spillage | 10 RMB/kWh |
| The cost of load shedding | 15 RMB/kWh |
| The maximum load shedding amount per unit time | 100 kWh/h |
| The maximum wind and solar spillage amount per unit time | 100 kWh/h |

The configured capacity of electrochemical energy storage is 51 GWh, and the configured capacity of hydrogen energy storage is 47 GWh.

Table 3 presents a comparison between existing research schemes and scenarios where marginal costs are not considered, as well as scenarios considering only a single type of energy storage, including hydrogen storage capacity, electrochemical storage capacity, and the profits obtained from system operations. It should be noted that profits are comprehensively considered based on the marginal costs of energy storage construction mentioned in this paper, with negative values indicating a decrease in profits compared to the scheme proposed in this paper. From the numerical values in the table, it can be inferred that considering marginal costs and the combination of different types of energy storage can improve the overall economic viability of the system while ensuring normal system operation.

As shown in Figure 6, the total shedding load throughout the year is 0, and the total wind and solar spillage throughout the year is 0.

Additionally, the long-term and short-term charging and discharging situations of electrochemical energy storage are shown in Figure 7, and the long-term and short-term charging and discharging situations of hydrogen energy storage are shown in Figure 8.

**Table 3.** Description of the cases.

| Case | Hydrogen Storage Capacity | Electrochemical Storage Capacity | Profit |
|---|---|---|---|
| Not considering marginal costs | 47 GWh | 52 GWh | −0.00126% |
| Only hydrogen storage | 70 GWh | 0 | −0.49282% |
| Only electrochemical storage | 0 | 75 GWh | −1.6436% |
| The case of this article | 46 GWh | 51 GWh | \ |

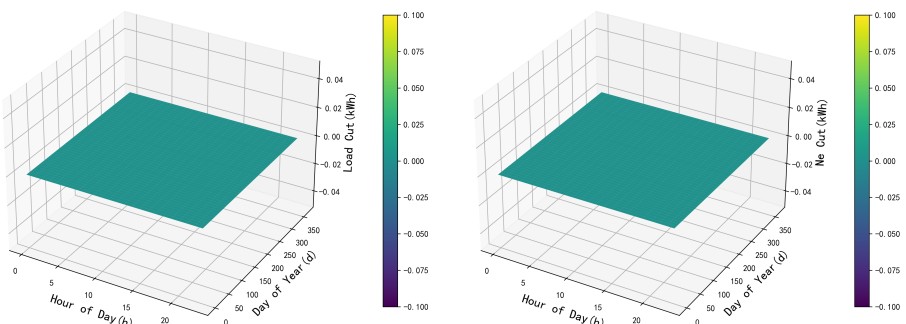

**Figure 6.** Wind and solar spillage and load shedding situation.

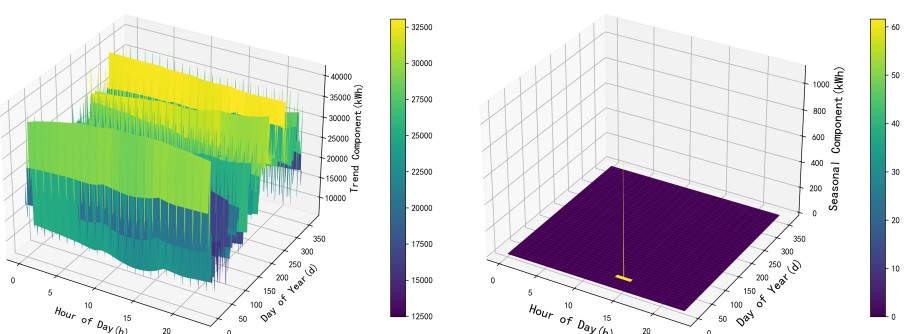

**Figure 7.** Hydrogen energy storage power output situation.

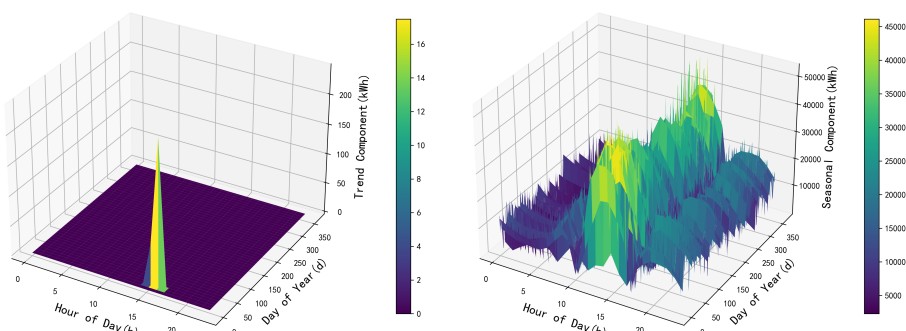

**Figure 8.** Electrochemical energy storage power output situation.

Comparing Figures 4 and 7, it can be observed that the long-term charging and discharging curve of hydrogen energy storage coincides with the long-term imbalance component of power. Comparing Figures 4 and 8, it can be observed that the short-term charging and discharging curve of electrochemical energy storage coincides with the seasonal imbalance component of power, which meets the expected situation.

As the existing renewable energy generation capacity cannot meet the requirements of the load operation, additional hydrogen energy is considered to achieve the balance of

power quantity in the system. The situation with additional hydrogen energy storage is illustrated in Figure 9.

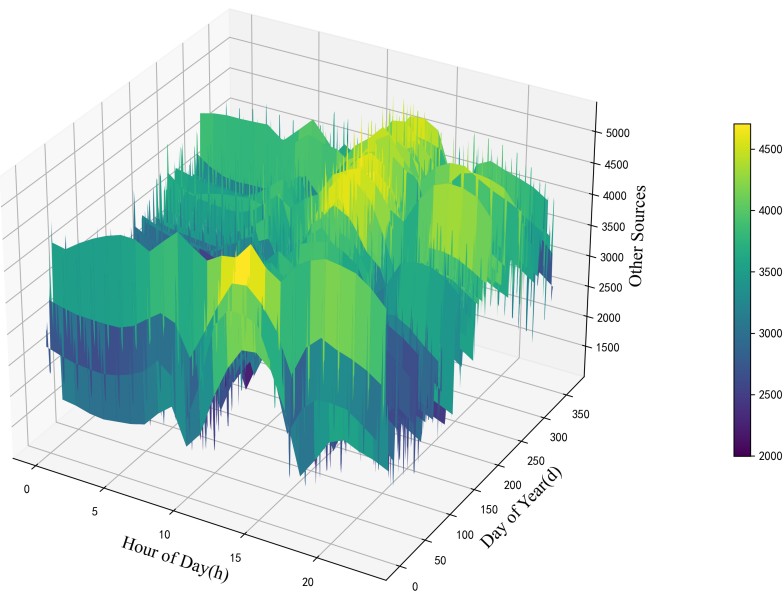

**Figure 9.** Externally provided hydrogen energy.

Selecting a day from the year and plotting the net load and energy storage output situation as shown in Figure 10, it can be observed from the graph that the power sources and loads achieve a balance of power throughout the day, with hydrogen energy mainly used to balance long-term power output and electrochemical energy storage used to balance short-term fluctuations in net load.

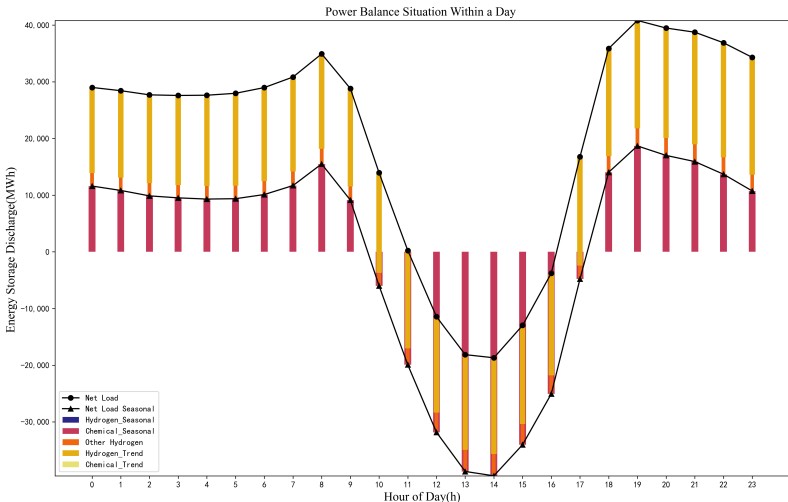

**Figure 10.** Overall power balance situation in the system.

## 5. Conclusions

To address the power system's electricity imbalance caused by the large-scale integration of new and fluctuating renewable energy sources, this paper proposes an energy storage planning method considering multi-time-scale electricity imbalance risks. The model captures the annual variations in renewable energy and load using decomposition methods to decouple the components of electricity imbalance over time. The decomposition results are then used as the basis for planning hydrogen and electrochemical energy storage capacities. By establishing an energy storage planning model to minimize overall costs, validated

with actual data, this paper makes the following contributions: Case studies based on operational data from a province in southern China demonstrate the effectiveness of the proposed model. This method achieves the multi-time-scale configuration of mixed energy sources, ensuring low-carbon, secure, and economically efficient operation of the power grid. By considering the marginal costs of energy storage construction and minimizing overall costs, the planning model increases the total benefits compared to not considering marginal costs. To enhance the energy utilization efficiency and leverage the advantages of different types of energy storage, this paper utilizes hydrogen storage, which does not experience self-discharge, to address trend-based fluctuations. Additionally, electrochemical energy storage, with rapid charge and discharge rates, is applied to handle short-term fluctuations in electricity. Compared to using a single type of energy storage, the overall benefits increase.

In the future, we will further consider the safety of hydrogen storage and incorporate considerations of the grid structure to determine the optimal locations and capacities for energy storage installations at various nodes, thus mitigating the impacts of large-scale renewable energy integration on the power system.

**Author Contributions:** Conceptualization, Q.L. and X.Z.; methodology, Y.Y.; validation, Q.H., G.W. and Y.H.; formal analysis, Y.L.; investigation, G.L.; resources, Q.L.; data curation, X.Z. All authors have read and agreed to the published version of the manuscript.

**Funding:** This work is supported by the Science and Technology Project of China Southern Power Grid Corporation. (Project No. 036000KK52220025 (GDKJXM20220329)).

**Data Availability Statement:** We are unable to publicly share the data supporting the reported results due to legal restrictions. However, upon request, we are willing to provide the data to qualified researchers under appropriate confidentiality agreements.

**Acknowledgments:** We would like to express our gratitude to all the reviewers for providing valuable feedback.

**Conflicts of Interest:** The authors declare no conflicts of interest.

## Abbreviation

The following abbreviation is used in this manuscript:

STL    Seasonal and Trend decomposition using Loess

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
