# Peer review of "Multi-Time-Scale Energy Storage Optimization Configuration for Power Balance in Distribution Systems"

_electronics, doi:10.3390/electronics13071379_

Round 1
Reviewer 1 Report
Comments and Suggestions for Authors
The article is written in good technical language. It contains a sufficient introduction to the topic, description of the methodology and results. The weak points of the article include the lack of descriptions of units in all charts from Fig. 3 to Fig. 9. They should be completed on the vertical axes, e.g. [kWh] or [W] etc.
Reviewer 2 Report
Comments and Suggestions for Authors
In this paper, the authors present a method for optimizing energy storage for low-carbon energy production seasonality.
The paper is very interesting, however, there is a lack of information that would help the reader. The authors should also present the source of some affirmations, for example, the increase of 600% in natural gas. The name of the province where the data was acquired is not mentioned, and the company responsible for obtaining the data is also not mentioned.
In the Case Study section, it was not clear how the costs were taken into account.
Also in the Case Study section, the results are not very well discussed, and the authors do not make it clear what they wanted to present. There are the figures with some results, but no discussion of the insights on it.
In the conclusion section, the authors should improve it, highlighting the benefits of their proposal and also considering the drawbacks of the study.
Reviewer 3 Report
Comments and Suggestions for Authors
Dear authors.
Please find attached our comments for your manuscript.

The manuscript need extensive proofreading in terms of language as there are grammatic and syntax errors, as well as occasionally incorrect vocabulary choices.
Reviewer 4 Report
Comments and Suggestions for Authors
The research paper „Multi-time-scale energy storage optimization configuration for power balance in distribution systems“ is convenient for the scope of the special issue“ Advances in Power System Dynamics, Stability, Control and Dispatch with Large-Scale Renewable Energy Penetrated“. The content of the paper is valuable from the theoretical and practical point of view. The main aim of research is clearly deffined. The title of scientific article is clear and it sufficiently reflects content. The abstract is informative. The references are appropriate.
In the manuscript are some irregularities which need corrections:
· The keywords should not be the same as those used in the title of the article.
· Titles of main chapters are not according to the template of journal.
· For equations, I recommend writing the units of quantities.
· Pictures in the Figure 4 are very small.
· A diagram should be drawn to clearly illustrate the overall solution procedure.
· In the conclusion, I recommend adding the main benefits of research for practice.
· Authors cited 30 references, which is not enough for this type of article.
GENERAL JUDGEMENT
The paper is acceptable for publication after minor revision
